# Immunogenicity and Safety of Homologous and Heterologous Prime-Boost of CoronaVac^®^ and ChAdOx1 nCoV-19 among Hemodialysis Patients: An Observational Prospective Cohort Study

**DOI:** 10.3390/vaccines11040715

**Published:** 2023-03-23

**Authors:** Phoom Narongkiatikhun, Kajohnsak Noppakun, Romanee Chaiwarith, Poramed Winichakoon, Surachet Vongsanim, Yuttitham Suteeka, Karn Pongsuwan, Prit Kusirisin, Nuttanun Wongsarikan, Kanda Fanhchaksai, Chantana Khamwan, Dararat Dankai, Vuddhidej Ophascharoensuk

**Affiliations:** 1Division of Nephrology, Department of Internal Medicine, Faculty of Medicine, Chiang Mai University, Chiang Mai 50200, Thailand; phoom.n@cmu.ac.th (P.N.); kajohnsak.noppakun@cmu.ac.th (K.N.); surachet.w@cmu.ac.th (S.V.); yuttitham.s@cmu.ac.th (Y.S.); karn.p@cmu.ac.th (K.P.); prit.kusirisin@cmu.ac.th (P.K.); 2Division of Infectious Diseases and Tropical Medicine, Department of Internal Medicine, Faculty of Medicine, Chiang Mai University, Chiang Mai 50200, Thailand; romanee.c@cmu.ac.th (R.C.); poramed.wi@cmu.ac.th (P.W.); 3Department of Internal Medicine, Faculty of Medicine, Chiang Mai University, Chiang Mai 50200, Thailand; nuttanun.w@cmu.ac.th; 4Division of Hematology and Oncology, Department of Pediatrics, Faculty of Medicine, Chiang Mai University, Chiang Mai 50200, Thailand; kanda.f@cmu.ac.th; 5Immunology Laboratory, Diagnostic Laboratory, Faculty of Medicine, Chiang Mai University, Chiang Mai 50200, Thailand; chantana.k@cmu.ac.th (C.K.); dararat.d@cmu.ac.th (D.D.)

**Keywords:** COVID-19, inactivated vaccine, replication-defective viral vectors vaccine, vaccine immunogenicity, hemodialysis

## Abstract

Background: Vaccines that prevent SARS-CoV-2 infection are considered the most promising approach to modulating the pandemic. There is scarce evidence on the efficacy and safety of different vaccine prime-boost combinations in MHD patients since most clinical trials have used homologous mRNA vaccine regimens. Methods: This prospective observational study assessed the immunogenicity and safety of homologous CoronaVac^®^ (SV-SV), ChAdOx1 nCoV-19 (AZD1222) (AZ-AZ), and the heterologous prime-boost of SV-AZ, among MHD patients. Results: A total of 130 MHD participants were recruited. On day 28, after the second dose, seroconversion results of the surrogate virus neutralization test were not different between vaccine regimens. The magnitude of the receptor-binding domain-specific IgG was highest among the SV-AZ. Different vaccine regimens had a distinct impact on seroconversion, for which the heterologous vaccine regimen demonstrated a higher probability of seroconversion (OR 10.12; *p* = 0.020, and OR 1.81; *p* = 0.437 for SV-AZ vs. SV-SV, and SV-AZ vs. AZ-AZ, respectively). There were no serious adverse events reported in any of the vaccine groups. Conclusions: Immunization with SV-SV, AZ-AZ, and SV-AZ could generate humoral immunity without any serious adverse events among MHD patients. Using the heterologous vaccine prime-boost seemed to be more efficacious in terms of inducing immunogenicity.

## 1. Introduction

Outbreaks of severe acute respiratory syndrome coronavirus-2 (SARS-CoV-2) infection have occurred in many countries throughout the world, subsequently resulting in a global pandemic [1]. The contagious and virulent profile of this virus has contributed to high rates of morbidity and mortality among patients who have been infected [2]. The risk of mortality from COVID-19 disease appears to be higher in patients aged over 70 years old, taking immunosuppressive agents, and those with pre-existing medical comorbidities, including chronic kidney disease (CKD) [3].

Vaccines that prevent SARS-CoV-2 infection are considered the most promising approach for modulating the pandemic and are being vigorously pursued. Many of these vaccines have been granted emergency use status through supporting evidence produced from standard hierarchically developmental studies [4]. Unfortunately, patients with kidney disease, including maintenance hemodialysis (MHD) patients, were generally excluded from those clinical trials [5,6,7]. In addition, there is scarce evidence of the efficacy and safety of different vaccine prime-boost combinations in MHD patients since most clinical trials have used homologous mRNA vaccine regimens. Therefore, data on the currently available vaccines against SARS-CoV-2 infection, in terms of efficacy and safety of various vaccine regimens among MHD patients, is still limited. 

Regarding Thailand’s Public Health Policy, the CoronaVac^®^ vaccine (Sinovac) and the ChAdOx1 nCoV-19 vaccine (AZD1222) have been selected as prioritized vaccines for high-risk groups, including MHD patients, during the initial phase of vaccine implementation in Thailand. Since the anticipated benefits of immunization have appeared to outweigh any potential health risks, immunization has been recommended for this specific group. Thus, this study aimed to evaluate the immunogenicity and safety of these specific SARS-CoV-2 vaccines among MHD patients.

## 2. Materials and Methods

### 2.1. Study Design and Participants

We conducted a single-center, observational prospective cohort study involving MHD patients between the 1st of June and the 31st of December 2021 at Maharaj Nakorn Chiang Mai Hospital, which is an affiliated hospital of Chiang Mai University, Thailand. Inclusion criteria identified MHD patients with a stable condition of at least three months before enrollment that were over the age of 18 and were able to provide informed consent. These participants were willing to receive one of the following vaccine regimens: homologous inactivated vaccine against SARS-CoV-2 regimen CoronaVac^®^ vaccine (Sinovac, Beijing, China) (SV-SV), homologous replication-defective viral vectors against SARS-CoV-2 regimen ChAdOx1 nCoV-19 vaccine (AZD1222) (AZ-AZ), or the heterologous prime-boost of inactivated vaccine followed by the replication-defective viral vectors vaccine (SV-AZ). Exclusion criteria eliminated patients who had previously been diagnosed with COVID-19 in the last 90 days, those presenting at a high-risk of epidemiology history within 14 days before enrollment, e.g., close contact with index cases or those visiting/living in an outbreak area, those who had received any other vaccine against SARS-CoV-2, those participating in another vaccine clinical trial, and those who had received any blood products, blood components, or immunoglobulin transfusions within the last 90 days. The exclusion criteria also eliminated women going through lactation, pregnancy, or planning a pregnancy during the study period, those patients presenting solid and/or hematological malignancy, and those diagnosed with confirmed Human Immunodeficiency Virus (HIV) infection.

### 2.2. Vaccine Regimen

#### 2.2.1. Homologous Inactivated Vaccine against SARS-CoV-2 Regimen CoronaVac^®^ Vaccine (Sinovac) (SV-SV)

Sinovac is an inactivated vaccine. It was developed in China. The vaccine was administered intramuscularly in two doses 28 days apart [8].

#### 2.2.2. Homologous Replication-Defective Viral Vectors against the SARS-CoV-2 Regimen ChAdOx1 nCoV-19 Vaccine (AZD1222) (AZ-AZ) 

This vaccine is based on a replication-incompetent chimpanzee adenovirus vector that expresses the spike protein. It is administered intramuscularly in two doses 12 weeks apart [9].

#### 2.2.3. Heterologous Prime-Boost of Inactivated Vaccine Followed by the Replication-Defective Viral Vectors Vaccine (SV-AZ)

This regimen has been applied since evidence was obtained from vaccine immunology demonstrating that the degree of immunogenicity in the heterologous prime-boost was higher than for the homologous-based regimen [10,11,12]. Furthermore, to cease the transmission of COVID-19 as soon as possible, a faster immunization regimen will be required, and the administration period will be shortened from the standard 12-week duration. Thus, this vaccine regimen would be administered intramuscularly, initially with SV, and followed by AZ 28 days apart.

#### 2.2.4. Data Collection, Laboratory Collection, and Immunogenicity Assessment

All eligible participants were assessed in terms of their general demographic data and hemodialysis data. The degree of immunogenicity of the SARS-CoV-2 vaccination and the results of the laboratory assessments, which would likely be associated with seroconversion rates, were collected at different specific time points depending on the vaccine regimen. For the SV-SV and SV-AZ regimens, blood was drawn one to three days before the first vaccination, one to three days before the second vaccination, and on day twenty-eight after the second vaccination. For the AZ-AZ regimen group, blood was drawn one to three days before the first vaccination, on day twenty-eight after the first vaccination, one to three days before the second vaccination, and on day twenty-eight after the second vaccination.

For the immunogenicity assessment, humoral immunity (HMI) was evaluated using the SARS-CoV-2 immunoglobulin G (IgG) assay, which assessed the antibodies against the S1 receptor-binding domain (RBD) of the SARS-CoV-2 spike protein and the results of the SARS-CoV-2 surrogate virus neutralization test (sVNT). 

### 2.3. SARS-CoV2 Anti-RBD IgG Assay

SARS-CoV-2 anti-RBD IgG antibodies were measured using the Abbott SARS-CoV-2 IgG II Quantification assay (Abbott Diagnostics, Abbott Park, IL, USA) performed on the Abbott Alinity instrument following the manufacturer’s instructions. This assay was a chemiluminescent microparticle immunoassay (CMIA) used for the quantitative determination of IgG antibodies to the SARS-CoV-2 spike protein in human serum. Accordingly, the results were reported in binding antibody units (BAU)/mL. A cutoff value of ≥7.1 BAU/mL was considered a positive result [13].

### 2.4. SARS-CoV-2 sVNT

The function of antibody percent inhibition was determined using the SARS-CoV-2 NeutraLISA surrogate neutralization assay (Euroimmun, Germany). The resulting percentage demonstrates the inhibitory ability of the producing antibody, which prevents the binding of RBD of viral SARS-CoV-2 S1 to the angiotensin-converting enzyme 2 (ACE2) receptor of human cells. A cutoff value of ≥35% inhibition (IH) was representative of seroconversion, while values ≥20–<35% IH and <20% IH were considered borderline and negative, respectively [14]. 

### 2.5. Safety Assessment and Adverse Events

A safety assessment was made by evaluating a self-recorded diary of adverse events (AEs) after the vaccines were administered. The diary consisted of checklists of the occurrence and severity of local reactions at the injection site (swelling, redness, and tenderness), systemic reactions (fever, headache, and fatigue), any other documented AEs occurring 14 days after vaccination, and an open field for any AEs that occurred during the period before the next round of vaccinations.

### 2.6. Study Outcomes

The primary endpoint was the seroconversion rate of sVNT on day 28 after completing the vaccination regimen. The secondary endpoints were the seroconversion rate of sVNT on day 28 after the first vaccination, the geometric mean titers (GMTs) of sVNT and RBD-specific IgG at specific time points, factors associated with seroconversion, and the AEs of each vaccine regimen.

### 2.7. Statistical Analysis

Descriptive data have been presented in numbers (%), mean ± SD values, and median (Interquartile range; IQR) values when appropriate. Comparisons of baseline characteristics and laboratory results were made between all cases. Each of the controls was determined using Chi-square and Fisher’s exact test, which were used to establish categorical data, while a Student’s *t*-test or Mann-Whitney U test were employed for continuous data, as was appropriate. A two-sided *p*-value of less than 0.05 was considered statistically significant. Factors associated with the seroconversion measured by sVNT after the complete course of each vaccine regimen were investigated by using backward stepwise logistic regression analyses. After univariate analysis, any variables indicating a *p*-value < 0.1, including all vaccine regimens, were employed in a multiple logistic regression analysis. All analyses were performed using Stata software version 16.0 (Stata Corp, College Station, TX, USA) [15].

## 3. Results

### 3.1. Baseline Clinical Characteristics and Laboratory Results of Participants

One hundred and thirty MHD participants, who had agreed to receive the COVID-19 vaccination, were recruited for this study. Among those, 89 (68.5%), 25 (19.2%), and 16 (12.3%) participants received the AZ-AZ, SV-AZ, and SV-SV regimens, respectively (Figure 1). There were 59 females (45.4%) with a mean age of 64.2 years old and a median hemodialysis vintage of 44 months. All baseline characteristics classified by the vaccine regimens were not determined to be different except for age, body mass index (BMI), and diabetes mellitus as an indication of comorbidity, which diverged across all groups (*p* < 0.05). The mean age of the SV-SV group was 49.8 years old, which was the youngest group when compared with the AZ-AZ and SV-AZ groups at 69.5 and 54.5 years old, respectively. The mean BMI value in the SV-SV group was the highest (27.9 kg/m^2^) across the three vaccine regimens, with a slight variability among members of the AZ-AZ (23.9 kg/m^2^) and the SV-AZ (22.7 kg/m^2^) groups. The percentage of patients with diabetes mellitus was the lowest in the SV-SV group (18.8%), followed by the SV-AZ (24.0%) and the AZ-AZ (53.9%) groups (Table 1).

### 3.2. SARS-CoV-2 Vaccine Immunogenicity

The seroconversion rates of sVNT on day 28 after a complete vaccination course were 88.0%, 78.7%, and 68.8% for SV-AZ, AZ-AZ, and SV-SV, respectively (*p* = 0.289) (Figure 2). The seroconversion rates of sVNT on day 28 after the first dose were 10.5%, 8.0%, and 0% for AZ-AZ, SV-AZ, and SV-SV, respectively (Appendix A, Appendix A). The rate of a positive result on day 28 after a complete course of vaccination assessed by the anti-spike RBD IgG antibodies also demonstrated that most MHD patients could generate a satisfactory humoral immune response at 88.0%, 87.5%, and 82.0% for SV-AZ, SV-SV, and AZ-AZ, respectively (*p* = 0.816) (Figure 2). On day 28, after a complete vaccination course, the median magnitude of anti-spike RBD IgG antibodies and sVNT was highest in SV-AZ (92.6% and 460.95 BAU/mL), followed by AZ-AZ (79.7% and 208.85 BAU/mL), and SV-SV (59.1%, and 139.34 BAU/mL) (Figure 3 and Figure 4). Notably, only the sVNT of SV-AZ was significantly higher than SV-SV (*p* = 0.031) (Figure 3), while the anti-spike RBD IgG antibodies of SV-AZ were significantly higher than SV-SV (*p* = 0.033) and AZ-AZ (*p* = 0.001) (Figure 4). 

### 3.3. Factors Associated with Seroconversion Measured by sVNT among MHD Patients

On day 28, after a complete vaccination course, 103 MHD patients (79.2%) developed seroconversion, as was determined by sVNT. Factors associated with seroconversion among MHD patients have been presented in Table 2 and Appendix A in the Appendix A. Age, plasma lymphocytes at the baseline, and the effects of different vaccine regimens of SV-AZ vs. SV-SV, and SV-AZ vs. AZ-AZ, were associated with seroconversion on univariate analysis. In the multivariate model, age, plasma lymphocytes > 16% at the baseline, SV-AZ vs. SV-SV, and AZ-AZ vs. SV-SV, were statistically significantly associated with seroconversion (Table 3). For every five-year increment of age and a plasma lymphocyte value > 16% at the baseline, the adjusted odds ratio values (OR) were 0.80 (95% CI 0.64–0.99; *p* = 0.047) and 6.47 (95% CI 2.15–19.46; *p* = 0.001), respectively. Comparisons made between the vaccine regimens indicated that the impact on the seroconversion rate was found to be different, of which SV-AZ vs. SV-SV, AZ-AZ vs. SV-SV, and SV-AZ vs. AZ-AZ demonstrated OR values of 10.12 (95% CI 1.45–70.57; *p* = 0.020), 5.58 (95% CI 1.16–26.75; *p* = 0.031), and 1.81 (95% CI 0.40–8.13; *p* = 0.437), respectively. Notably, the heterologous vaccine regimen demonstrated a higher degree of probability of seroconversion.

### 3.4. Safety and Adverse Events after Vaccination

There were no serious AEs, either local or systemic, reported in any of the vaccine groups. The number of AEs after the second dose of vaccination were greater than after the first dose in all vaccine regimens except for the AZ-AZ group. The most common AE was a local reaction, which was mainly indicated by tenderness at the injection site. Concordantly, the degree of tenderness was higher after the second dose of vaccination than after the first dose across all vaccine regimens except for the AZ-AZ group. Among the three vaccine regimens, SV-AZ had the most frequent reports of AEs. Table 4 demonstrates the presence of any adverse events over the course of vaccination administration during each vaccine regimen.

## 4. Discussion

Since the risk of mortality from COVID-19 disease is known to be higher among MHD patients when compared with the general population, the implementation of preventive strategies using effective vaccines would be crucial. However, most clinical studies involving vaccines have generally excluded this group of patients. In this study, we investigated the degree of vaccine immunogenicity of SV-SV, AZ-AZ, and SV-AZ regimens among 130 MHD patients. On day 28, after a complete vaccination course, most MHD patients could develop seroconversion without any serious AEs. The rate of seroconversion between each vaccine regimen was not determined to be different. 

Among CKD patients, including MHD patients, the degree of vaccine immunogenicity was diminished after the presence of defects during the immune response [16]. In our study, after administering the homologous inactivated vaccine, homologous replication-defective viral vectors vaccine, and the heterologous prime-boost of inactivated and replication-defective viral vectors vaccine, there were 68.8%, 78.7%, and 88.0% rates of seroconversion, and 87.5%, 82.0%, and 88.0% positive results for the anti-spike RBD IgG antibodies, respectively. These results were similar to those of the previous COVID-19 vaccine immunogenicity studies involving MHD patients, mainly on mRNA platforms, which demonstrated seroconversion rates ranging from 72.8% to 96.4% [17]. This indicated that other vaccine platforms, in addition to the mRNA vaccine, could generate a satisfactory immune response among MHD patients. 

Although an immunological correlation of protection (COP) against clinical outcomes after various COVID-19 vaccinations is still not completely understood, increasing amounts of evidence have revealed that a range of immune measures, such as sVNT and anti-spike RBD IgG, were directly correlated to the COP [18]. We found that on day 28, after a complete vaccination course, the magnitude of sVNT and anti-spike RBD IgG were substantially increased in all vaccine regimens. Both parameters were highest among SV-AZ (92.6% and 460.95 BAU/mL), followed by AZ-AZ (79.7% and 208.85 BAU/mL) and SV-SV (59.1% and 139.34 BAU/mL). However, those anti-spike RBD IgG levels were lower than the cutoff value of the anti-spike RBD IgG of 506 BAU/mL following administration of AZ-AZ, which had been associated with 80% vaccine efficacy against symptomatic infections in the previous report [18]. Although this reference value could not directly be applied to all vaccine regimens, this extrapolation led to the consideration that an additional booster dose might be needed for our MHD patients.

Several factors that correlated to COVID-19 vaccine immunogenicity among MHD patients have been identified. Akyol et al. reported that a younger age [19], a body mass index below 30 kg/m^2^ [19], lower intravenous iron supplementation [19], a hepatitis B vaccine antibody response > 20 IU/mL [20,21], higher serum albumin [19,21], and a higher Kt/Vurea value [21] were all correlated with greater seroconversion rates [17]. Nonetheless, an older age [21,22,23,24], those currently undergoing immunosuppressive treatment, lower dialysis vintage, lower serum albumin, lower white blood cell or lymphocyte counts, and lower hemoglobin levels were all correlated to lower seroconversion rates [17,25]. In our study, every 5-year increment of age and plasma lymphocytes ≤ 16% at the baseline were associated with a lower probability of an effective seroconversion rate after accounting for other factors. These results were consistent with the outcomes of other previous studies. In addition, our results showed the threshold of plasma lymphocyte percentage which have an impact on the rate of seroconversion. Even though serum albumin, Kt/Vurea values, dialysis vintage, and hemoglobin levels demonstrated similar impacts on the seroconversion rate as had been reported in previous publications, those variables did not reach statistical significance in this study. This may have been due to the relatively small sample size employed in this study. 

Different vaccine platforms were one of the factors that had an impact on the degree of immunogenicity. In this study, the magnitude of sVNT among those receiving the adenovirus-based vaccine regimen was greater than for those receiving the inactivated vaccine regimen. Multivariate analysis also demonstrated that the adenovirus-based vaccine regimen was associated with a higher degree of probability for seroconversion when compared with those receiving the inactivated vaccine regimen. This finding correlated with a systematic review and network meta-analysis of the COVID-19 vaccines among the general population, which indicated that the mRNA vaccine could promote the highest neutralizing antibody response followed by the adenovirus-based vaccine, the pro-subunit vaccine and the inactivated vaccine, respectively [26]. Therefore, the effect of the different vaccine platforms on immunogenicity among MHD patients might be the same as for the general population.

During the COVID-19 pandemic, providing the homologous vaccine regimen may not always be possible since there may be a lack of availability of the vaccine and an inability to predict what would be considered an adequate supply. In addition, some vaccine regimens require long intervals between the first and second shots to promote an effective degree of immunogenicity, which may not be attainable when attempting to overtake the dispersion of COVID-19. Growing amounts of evidence have indicated that the heterologous prime-boost regimen could promote a higher degree of immunogenicity during a shorter time interval between injections under the safety profile [27,28,29]. Consistent with previous findings, our study found that the heterologous prime-boost regimen of SV-AZ exhibits the highest magnitude of both anti-spike RBD IgG antibodies and sVNT. Furthermore, the multivariate analysis indicated that the probability of seroconversion was more apparent in the heterologous prime-boost regimen when compared with the homologous regimen. The underlying mechanism of the immunological enhancer of different vaccine platforms might involve the induction of the immune system through different pathways [30]. We have anticipated that this scientific vaccine strategy should be further evaluated since it might be indicative of a necessary protocol for other populations who are prone to having an inadequate immunological response. 

This study evaluated the safety profile and the AEs of different vaccine regimens among MHD patients. We found that there were no serious AEs reported in any of the vaccine groups. The most commonly reported AEs involved a local reaction, which was mainly indicated by tenderness at the injection site. Our results were similar to most of the previous studies of each vaccine regimen among the general population [8,31]. These results demonstrated that SV-SV, AZ-AZ, and SV-AZ regimens were tolerable and generally safe among MHD patients. Notably, the frequency of the reported vaccine side effects and the severity of those events were higher after the second dose in members of each vaccine group, especially among members receiving the heterologous prime-boost regimen. In correlation with the previously recorded vaccine immunogenicity results, it could be concluded that a greater degree of reactogenicity would result in a higher degree of immunogenicity. Interestingly, this proposed rationale was consistent with the results of a previous systemic review [32]. However, additional data will be required to confirm this assumption.

Unlike the determinations of other previously published studies, this study evaluated various vaccine platforms other than that of the mRNA-based platform and involved diverse vaccine regimens, including the administration of both homologous and heterologous regimens among MHD patients. In addition, our participants were truly naïve to COVID-19 infection since a lack of preformed antibodies was detected at the baseline before patients received the vaccination. This contributed to the reliability of the study outcomes as the effect of the immunogenicity was mainly from the vaccine and not from a natural infection. Nevertheless, our study did have some limitations, including the fact that there were a relatively small number of MHD patients included in each vaccine regimen and a lack of healthy controls. Furthermore, the effect of each vaccine regimen on other variants of concern was not determined. We also did not evaluate the cell-mediated immune (CMI) response, which was one of the important factors for assessing immunogenicity.

## 5. Conclusions

In conclusion, both a homologous or heterologous prime-boost of either inactivated vaccine or replication-defective viral vectors vaccine against SARS-CoV2 could generate a humoral immune response without any serious adverse events among MHD patients. Prescribing different vaccine platforms seemed to be more efficacious in terms of inducing vaccine immunogenicity and might be the solution for an individual with a poor immune response or for someone who is immunocompromised. Further studies on the heterologous vaccine prime-boost vaccine regimen and additional booster doses with different vaccine platforms are warranted.

## Figures and Tables

**Figure 1 vaccines-11-00715-f001:**
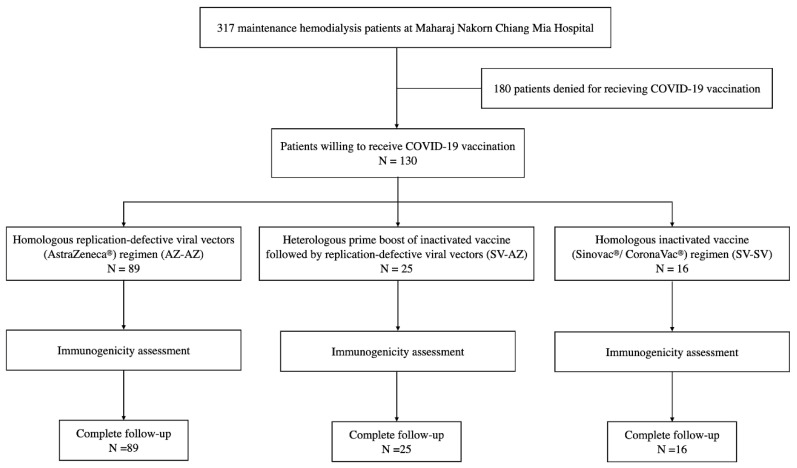
Study design and flow diagram. AZ-AZ, homologous AZD1222 regimen; SV-AZ, heterologous Sinovac-AZD1222 regimen; SV-SV, homologous Sinovac regimen.

**Figure 2 vaccines-11-00715-f002:**
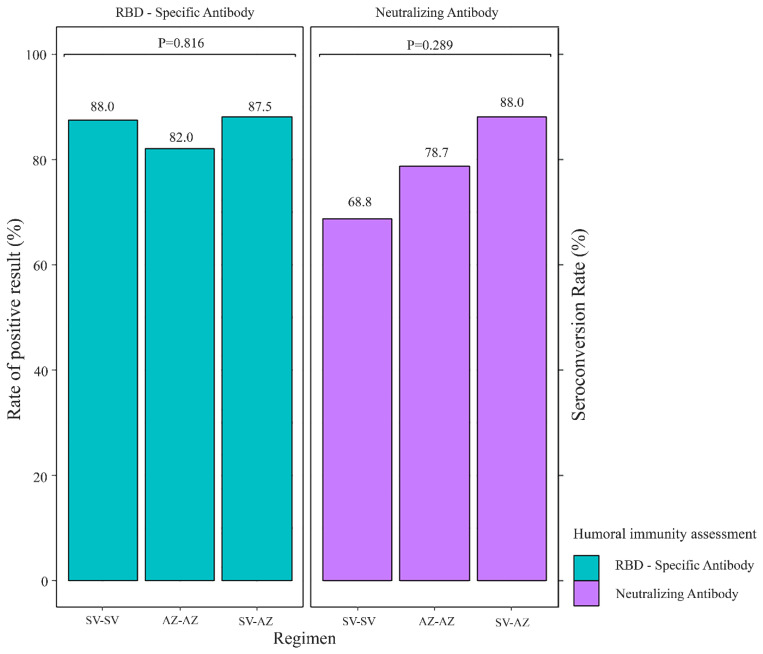
Rate of positive results by anti-spike RBD-IgG antibodies and seroconversion rates assessed by SARS-CoV-2 surrogate virus neutralization test on day 28 after the complete course of each vaccine regimen among MHD patients.AZ-AZ, homologous AZD1222 regimen; SV-AZ, heterologous Sinovac-AZD1222 regimen; SV-SV, homologous Sinovac regimen.

**Figure 3 vaccines-11-00715-f003:**
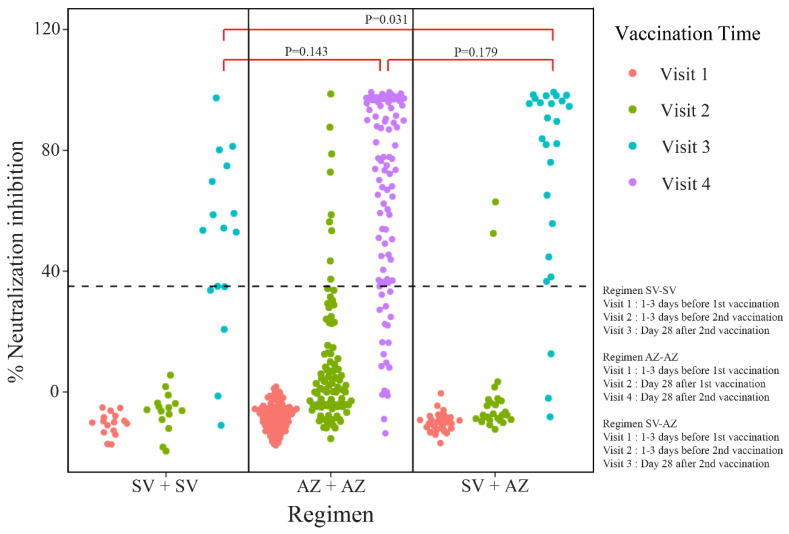
Percentage of SARS-CoV-2 surrogate virus neutralization test results at different visits of each vaccine regimen. AZ-AZ, homologous AZD1222 regimen; SV-AZ, heterologous Sinovac-AZD1222 regimen; SV-SV, homologous Sinovac regimen.

**Figure 4 vaccines-11-00715-f004:**
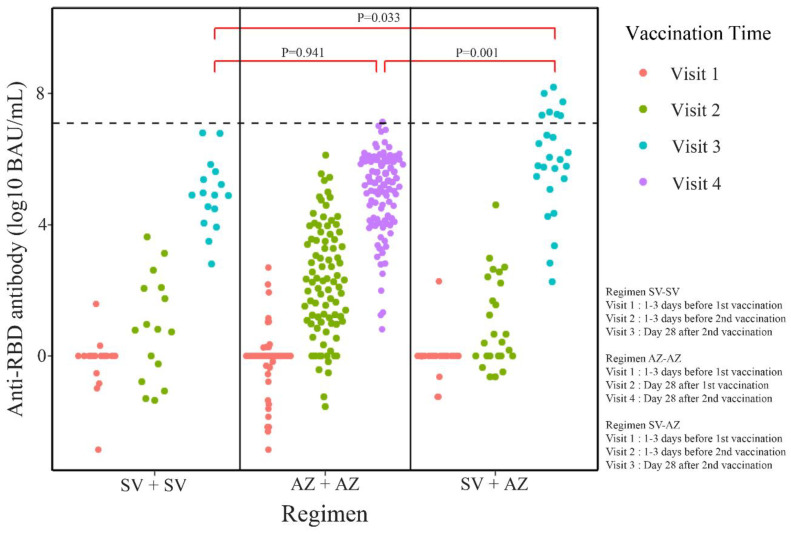
Titer of anti-spike RBD IgG antibodies at different visits of each vaccine regimen. AZ-AZ, homologous AZD1222 regimen; SV-AZ, heterologous Sinovac-AZD1222 regimen; SV-SV, homologous Sinovac regimen.

**Table 1 vaccines-11-00715-t001:** Baseline clinical characteristics and laboratory results of participants according to the vaccine regimen.

Characteristic	Total (N = 130)	SV-SV (N = 16)	AZ-AZ (N = 89)	SV-AZ (N = 25)	*p* Value
Age—year	64.2 ± 14.3	49.8 ± 10.2	69.5 ± 10.2	54.5 ± 17.7	<0.001
Female	59 (45.3)	5 (31.3)	41 (46.1)	13 (52.0)	0.417
Body mass index—kg/m^2^	24.1 ± 5.4	27.9 ± 8.8	23.9 ± 4.4	22.7 ± 5.1	0.007
Comorbid disease					
Hypertension	124 (95.4)	16 (100.0)	84 (94.4)	24 (96.0)	1.000
Dyslipidemia	53 (40.8)	6 (37.5)	36 (40.5)	11 (44.0)	0.913
Diabetes mellitus	57 (43.9)	3 (18.8)	48 (53.9)	6 (24.0)	0.003
Cardiovascular disease (MI, HF)	25 (19.2)	3 (18.8)	17 (19.1)	5 (20.0)	1.000
Cerebrovascular disease	4 (3.1)	1 (6.3)	3 (3.4)	0 (0)	0.535
Chronic obstructive pulmonary disease	2 (1.5)	0 (0)	1 (1.1)	1 (4.0)	0.533
Connective tissue disease	8 (6.2)	1 (6.3)	3 (3.4)	4 (16.0)	0.048
Liver disease	6 (4.6)	0 (0)	5 (5.6)	1 (4.0)	1.000
Cause of end-stage kidney disease					0.010
Diabetic kidney disease	54 (41.5)	2 (12.5)	46 (51.7)	6 (24.0)	
Hypertensive nephropathy	18 (13.9)	2 (12.5)	14 (15.7)	2 (8.0)	
Obstructive uropathy	9 (6.9)	2 (12.5)	4 (4.5)	3 (12.0)	
Glomerular disease	28 (21.5)	7 (43.8)	13 (14.6)	8 (32.0)	
Others causes	6 (4.6)	1 (6.3)	3 (3.4)	2 (8.0)	
Unknown cause	15 (11.5)	2 (12.5)	9 (10.1)	4 (16.0)	
Dialysis vintage—months (min, max)	44 (26, 70)	32.5 (23, 69.5)	45 (30, 69)	32 (17, 75)	0.812
Dialysis schedule					0.802
1–2 time/week	25 (19.2)	4 (25.0)	16 (18.0)	5 (20.0)	
3–4 time/week	105 (80.8)	12 (75.0)	73 (82.0)	20 (80.0)	
Mode of hemodialysis					0.582
Conventional hemodialysis	119 (91.5)	14 (87.5)	83 (93.3)	22 (88.0)	
Online hemodiafiltration	11 (8.5)	2 (12.5)	6 (6.7)	3 (12.0)	
Presence of urine output	83 (63.9)	10 (62.5)	56 (62.9)	17 (68.0)	0.890
<200 mL	28 (33.7)	3 (30.0)	19 (33.9)	6 (35.3)	1.000
≥200 mL	55 (66.3)	7 (70.0)	37 (66.1)	11 (64.7)	
Iron supplement	82 (63.1)	12 (75.0)	57 (64.0)	13 (52.0)	0.346
Intravenously	68 (82.9)	10 (83.3)	46 (80.7)	12 (92.3)	0.743
Oral	14 (17.1)	2 (16.67)	11 (19.3)	1 (7.7)	
Laboratory result					
Hemoglobin—g/dL	10.3 ± 1.4	10.3 ± 1.0	10.3 ± 1.34	10.4 ± 1.7	0.965
White blood cells—cells/mm^3^	6738.5 ± 2160.7	6970.6 ± 1852.9	6520.7 ± 2054.9	7365.6 ± 2615.7	0.203
Polymorphonuclear cell—%	63.8 ± 9.5	64.1 ± 8.5	63.5 ± 9.0	64.6 ± 11.6	0.889
Lymphocyte—%	22.1 ± 6.4	21.6 ± 3.9	22.1 ± 6.3	22.0 ± 7.9	0.959
≤16	21 (16.2)	1 (6.3)	14 (15.7)	6 (24.0)	0.322
>16	109 (83.9)	15 (93.8)	75 (84.3)	19 (76.0)	
Platelet—×10^3^ cells/mm^3^	219 (175, 261)	215 (187, 246)	216 (179, 259)	239 (156, 278)	0.789
Transferrin saturation—% (min, max)	31 (24, 42)	31 (19.5, 52.5)	30 (24, 39)	35 (27, 47)	0.079
Ferritin—µg/L	568.9 ± 414.4	478.56 ± 271.01	564.1 ± 436.4	643.8 ± 409.2	0.455
C-reactive protein—mg/L	3.1 (3.12, 6.2)	3.12 (3.12, 6)	3.1 (3.1, 6.5)	3.1 (3.1, 4.6)	0.785
Albumin—g/dL	4 (3.8, 4.2)	4.1 (3.85, 4.4)	4 (3.8, 4.2)	4.1 (3.9, 4.2)	0.122
Kt/v	1.6 ± 0.5	1.49 ± 0.32	1.6 ± 0.5	1.68 ± 0.52	0.329

All values are presented in number (%) unless otherwise specified. AZ-AZ, homologous AZD1222 regimen; HF, heart failure; MI, myocardial infarction; SV-AZ, heterologous Sinovac-AZD1222 regimen; SV-SV, homologous Sinovac regime.

**Table 2 vaccines-11-00715-t002:** Univariate analysis of factors associated with the seroconversion measured by sVNT among maintenance hemodialysis patients.

Variable	Positive	Univariate Analysis		
		OR	95%CI	*p* Value
No.	103/130 (79.2)			
Age—(every 5-year increment in age)	63.2 ± 14.4	0.81	0.66–0.99	0.040
Female	48/59 (81.4)	1.27	0.54–3.00	0.587
Vaccine regimen				
SV-AZ vs. SV-SV		3.33	1.49–7.43	0.003
AZ-AZ vs. SV-SV		1.67	0.93–3.01	0.085
SV-AZ vs. AZ-AZ		1.99	1.03–3.83	0.039
Body mass index—kg/m^2^	23.8 ± 4.4	0.95	0.88–1.02	0.149
Comorbid disease				
Diabetes mellitus	42/57 (73.7)	0.55	0.23–1.29	0.172
Connective tissue disease	6/8 (75.0)	0.77	0.15–4.06	0.761
Dialysis vintage—months	44 (27, 70)	1.00	0.99–1.01	0.696
Laboratory result				
Hemoglobin—g/dL	10.38 ± 1.44	1.12	0.83–1.51	0.461
White blood cell—cells/mm^3^	6654.0 ± 2077.6	1.00	1.00–1.00	0.385
Lymphocyte—%	22.7 ± 6.1	1.08	1.01–1.17	0.031
≤16	11 (10.7)	Ref.		
>16	92 (89.3)	4.92	1.81–13.38	0.002
Total Lymphocyte	1462.8 ± 480.1	1.00	1.00–1.00	0.210
Ferritin—µg/L	460 (302, 724)	1.00	1.00–1.00	0.796
C-reactive protein—mg/L	3.12 (3.12, 7.7)	1.07	0.95–1.17	0.290
Albumin—g/dL	4 (3.9, 4.2)	1.91	0.57–6.37	0.292
Kt/v	1.6 ± 0.5	1.94	0.73–5.15	0.182

All values are presented in number (%) unless otherwise specified. AZ-AZ, homologous AZD1222 regimen; SV-AZ, heterologous Sinovac-AZD1222 regimen; SV-SV, homologous Sinovac regimen.

**Table 3 vaccines-11-00715-t003:** Multivariate analysis of factors associated with the seroconversion measured by sVNT among maintenance hemodialysis patients.

Variable	Positive	Multivariate Analysis		
		Adjusted OR	95%CI	*p* Value
No.	103/130 (79.2)			
Age—(every 5-year increment in age)	63.2 ± 14.4	0.80	0.64–0.99	0.047
Vaccine regimen (Regimen vs. Ref.)				
SV-AZ vs. SV-SV		10.12	1.45–70.57	0.020
AZ-AZ vs. SV-SV		5.58	1.16–26.75	0.031
SV-AZ vs. AZ-AZ		1.81	0.40–8.13	0.437
Lymphocyte—%	22.7 ± 6.1			
≤16	11 (10.7)	Ref.		
>16	92 (89.3)	6.47	2.15–19.46	0.001

All values are presented in number (%) unless otherwise specified. AZ-AZ, homologous AZD1222 regimen; SV-AZ, heterologous Sinovac-AZD1222 regimen; SV-SV, homologous Sinovac regimen.

**Table 4 vaccines-11-00715-t004:** Adverse events during the course of vaccination administration for each vaccine regimen.

Side Effects and AE	Regimen					
	SV-SV (N = 16)		SV-AZ (N = 25)		AZ-AZ (N = 89)	
	After Dose 1	After Dose 2	After Dose 1	After Dose 2	After Dose 1	After Dose 2
Event	4/16 (25.0)	7/16 (43.8)	16/25 (64.0)	20/25 (80.0)	46/89 (51.7)	25/89 (28.1)
Local reaction	3 (18.8)	5 (31.3)	10 (40.0)	18 (72.0)	25 (28.1)	10 (11.2)
Swelling	0	0	0	1 (4.0)	0	1 (1.1)
Redness	0	0	0	1 (4.0)	0	1 (1.1)
Tenderness	3 (100.0)	5 (100.0)	10 (100.0)	18 (100.0)	25 (100.0)	10 (100.0)
Grade 1	3 (100.0)	3 (60.0)	8 (80.0)	5 (27.8)	16 (64.0)	8 (80.0)
Grade 2	0	2 (40.0)	2 (20.0)	11 (61.1)	9 (36.0)	1 (10.0)
Grade 3	0	0	0	2 (11.1)	0	1 (10.0)
Fever	3 (12.0)	6 (24.0)	11 (12.4)	1 (1.1)	0	1 (6.3)
Headache	4 (16.0)	5 (20.0)	9 (10.1)	7 (7.9)	1 (6.3)	0
Fatigue	5 (20.0)	6 (24.0)	12 (13.5)	3 (3.4)	0	0
Rash	1 (6.3)	1 (6.3)	0	1 (4.0)	0	0
Abdominal pain	0	0	0	1 (4.0)	1 (1.1)	0
Others	1 (6.3)	1 (6.3)	5 (20.0)	3 (12.0)	14 (15.7)	9 (10.1)

## Data Availability

The data that support the findings of this study are available from the corresponding author, V.O., vuddhidej.o@cmu.ac.th, upon reasonable request. The data are not publicly available due to privacy or ethical restrictions.

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
