# Peer review of "Immunogenicity and Safety of Homologous and Heterologous Prime-Boost of CoronaVac® and ChAdOx1 nCoV-19 among Hemodialysis Patients: An Observational Prospective Cohort Study"

_vaccines, 2023, doi:10.3390/vaccines11040715_

Round 1

Reviewer 1 Report

Narongkiatikhun and colleagues have examined the antigenicity of three different vaccine regimens on hemodialysis patients.  A number of studies on COVID vaccination and hemodialysis patients have been published, but few of these studies have examined the response to CoronaVac and none in the context of heterologous prime-boost. 

Major Comments

1.  Lines 78-86:  Please re-word.  At the start of this section, the authors state that individuals who had been previously vaccinated against COVID were excluded (line 82).  Toward the end of this section they write that individuals who had received a vaccine in the last 14 days were excluded (line 86).  It's not clear what criteria were used, though Figure 1 makes me think line 82 is correct.  See also lines 343-345.  

2.  Lines 167-168:  The SV-SV group is considerable smaller than the other two.  However, given the emphasis on heterologous prime-boost, I think this can be overlooked, assuming all the statistical tests confirm the reported outcomes.

3.  Figures 3 & 4:  Data for the AZ+AZ group is show for four visits.  The blood draw schedule is explained on lines 114-118, and it's clear that visit four for the AZ+AZ group is equivalent to visit three for the other groups.  Regardless, showing data from four visits for AZ+AZ creates some confusion.  If data from the AZ+AZ third visit don't contribute to the overall conclusion of the paper, I suggest removing it from the figures.  Statistics shown only compare results for final visits.

Author Response

Quincy Zhang

Editor-in-Chief

Vaccines

Re: “Immunogenicity and Safety of Homologous and Heterologous Prime-Boost of CoronaVac® and ChAdOx1 nCoV-19 Among Hemodialysis Patients: An Observational Prospective Cohort Study” (Reference number: Vaccines-2243165)  

March 13rd, 2023

Dear Editors of Vaccines

Enclosed please find the revised manuscript entitled “Immunogenicity and Safety of Homologous and Heterologous Prime-Boost of CoronaVac® and ChAdOx1 nCoV-19 Among Hemodialysis Patients: An Observational Prospective Cohort Study” for consideration for publication in the Vaccines. We would like to thank the editors and the reviewers for your valuable suggestions and the opportunity to revise and improve our manuscript.

We have revised our manuscript in accordance with the comments and suggestions made. Please find the responses to the reviewer and the revisions made to the manuscript below.

Responses to the Editor and the Reviewers

We would like to thank the editor and reviewers for the consideration of our clinical study.  We have revised our manuscript according to the reviewers’ suggestions.  We believe that the revised manuscript has been greatly improved after this revision and that it is suitable for its publication in the MDPI.

Responses to Reviewer #1

Lines 78-86:  Please re-word.  At the start of this section, the authors state that individuals who had been previously vaccinated against COVID were excluded (line 82).  Toward the end of this section they write that individuals who had received a vaccine in the last 14 days were excluded (line 86).  It's not clear what criteria were used, though Figure 1 makes me think line 82 is correct.  See also lines 343-345.  

Answer: We would like to thank the reviewer for the constructive comments. We have removed the content “those who had received an attenuated live vaccine in the past 28 days, those who had received an inactivated or subunit vaccine in the last 14 days” as suggested by the reviewer.

“The exclusion criteria also eliminated women going through lactation, pregnancy, or a planned pregnancy during the study period, those patients presenting solid and/or hematological malignancy, and those diagnosed with confirmed Human Immunodeficiency Virus (HIV) infection.”

Materials and Methods
Study Design and Participants
Page 2, paragraph 4, line 87-90

Lines 167-168:  The SV-SV group is considerable smaller than the other two.  However, given the emphasis on heterologous prime-boost, I think this can be overlooked, assuming all the statistical tests confirm the reported outcomes.

Answer: We would like to thank the reviewer for your suggestion. We have added the statistical tests comparing the rate of positive result of RBD-specific antibody and the seroconversion rate between each vaccine regimen, which are our interested outcomes. Nonetheless, those results were not different with our primary analysis. We added the Table S3 in the supplementary data.

Table S3. Comparison rate of the positive result of RBD-specific antibody and the seroconversion rate between each vaccine regimen.

Antibody

SV+SV (N=16)

SV+AZ (N=25)

p-value

SV+SV (N=16)

AZ+AZ (N=89)

p-value

SV+AZ (N=25)

AZ+AZ (N=89)

p-value

RBD

1.000

0.733

0.762

Negative

2 (12.50)

3 (12.00)

2 (12.50)

16 (17.98)

3 (12.00)

16 (17.98)

Positive

14 (87.50)

22 (88.00)

14 (87.50)

73 (82.02)

22 (88.00)

73 (82.02)

Neutralize

0.100

0.399

0.501

Negative

3 (12.00)

2 (12.50)

2 (12.50)

12 (13.48)

3 (12.00)

12 (13.48)

Borderline

0

3 (18.75)

3 (18.75)

7 (7.87)

0

7(7.87)

Positive

22 (88.00)

11 (68.75)

11 (68.75)

70 (78.65)

22 (88.00)

70 (78.65)

Figures 3 & 4:  Data for the AZ+AZ group is show for four visits.  The blood draw schedule is explained on lines 114-118, and it's clear that visit four for the AZ+AZ group is equivalent to visit three for the other groups.  Regardless, showing data from four visits for AZ+AZ creates some confusion.  If data from the AZ+AZ third visit don't contribute to the overall conclusion of the paper, I suggest removing it from the figures.  Statistics shown only compare results for final visits.

Answer: We would like to thank the reviewer for the constructive comments. We have removed the third visit of AZ+AZ from figure 3, and figure 4 as suggested by the reviewer.

Figure 3. Percentage of SARS-CoV-2 surrogate virus neutralization test results at different visits of each vaccine regimen.

Figure 4. Titer of anti-spike RBD IgG antibodies at different visits of each vaccine regimen.

Thank you again for all the assistance given by both editor and reviewers. Following extensive amendments in line with your very helpful comments we wish to resubmit out work for your consideration for publication in the MDPI Vaccines.

We look forward to your response.

Yours sincerely,

Sincerely yours,

Vuddhidej Ophascharoensuk (signed)

Vuddhidej Ophascharoensuk, MD.

Division of Nephrology, Department of Internal medicine, Faculty of Medicine,

Chiang Mai University, Chiang Mai, Thailand.

Tel. +66 53 935482    Fax: +66 53935481    

Email: vuddhidej.o@cmu.ac.th

ORCID: 0000-0002-6041-7989

  1. Authors and affiliations:

-Please replaced Dr. Karn Pongsuwan, M.D. to Dr. Karn Pogsuwan, M.D.

-Please changed the E-mail address of Dr. Romanee Chaiwarith, M.D. (E-mail: romanee.c@cmu.ac.th) and Dr. Poramed Winichakoon, M.D. (E-mail: poramed.wi@cmu.ac.th)

  1. Word count: 201 words (Abstract), 3,364 words (Introduction to Conclusion), 16 pages, 4 Figures, 4 Tables, 3 Supplementary Tables, and 33 References.

Reviewer 2 Report

The two key statements to support the current manuscript are 1/ "It is unknown as to whether those vaccines are as effective in maintenance hemodialysis (MHD) patients as they have been found to be in healthy people." and 2/ "However, data on the currently available vaccines against SARS-CoV-2 infection in terms of efficacy and safety among MHD patients, is still limited."  These statements are not entirely accurate.  There have been several published papers attesting to the fact that a variety of SARS-CoV-2 vaccines, including a similar heterologous combination to one of those studied here, are as effective in MHD patients as in healthy controls.  Moreover this manuscript does not address this question in not having healthy controls as a comparator.  Safety and efficacy of different vaccines and vaccine combinations has been covered in the published literature and these various studies should have been referenced.  What sets this aside is the comparison of different vaccine sequences, which is covered more extensively here than in other publications.  Assuming that there are no differences in vaccine sequences due for clinical reasons, the data could have value if there were a healthy comparator group and adequate rationale for studying the different vaccine combinations were provided.

Author Response

Quincy Zhang

Editor-in-Chief

Vaccines

Re: “Immunogenicity and Safety of Homologous and Heterologous Prime-Boost of CoronaVac® and ChAdOx1 nCoV-19 Among Hemodialysis Patients: An Observational Prospective Cohort Study” (Reference number: Vaccines-2243165)  

March 13rd, 2023

Dear Editors of Vaccines

Enclosed please find the revised manuscript entitled “Immunogenicity and Safety of Homologous and Heterologous Prime-Boost of CoronaVac® and ChAdOx1 nCoV-19 Among Hemodialysis Patients: An Observational Prospective Cohort Study” for consideration for publication in the Vaccines. We would like to thank the editors and the reviewers for your valuable suggestions and the opportunity to revise and improve our manuscript.

We have revised our manuscript in accordance with the comments and suggestions made. Please find the responses to the reviewer and the revisions made to the manuscript below.

Responses to the Editor and the Reviewers

We would like to thank the editor and reviewers for the consideration of our clinical study.  We have revised our manuscript according to the reviewers’ suggestions.  We believe that the revised manuscript has been greatly improved after this revision and that it is suitable for its publication in the MDPI.

Responses to Reviewer #2

We would like to thank the reviewer for the constructive comments.  Below are our point-by-point responses to the reviser’s suggestions.

The two key statements to support the current manuscript are 1/ "It is unknown as to whether those vaccines are as effective in maintenance hemodialysis (MHD) patients as they have been found to be in healthy people." and 2/ "However, data on the currently available vaccines against SARS-CoV-2 infection in terms of efficacy and safety among MHD patients, is still limited."  These statements are not entirely accurate.  There have been several published papers attesting to the fact that a variety of SARS-CoV-2 vaccines, including a similar heterologous combination to one of those studied here, are as effective in MHD patients as in healthy controls.

Answer: We agree that those two key statements are not entirely accurate as you noted since there are several publications of SARS-CoV-2 vaccines among MHD patients. Nevertheless, most of the publications are mainly homologous mRNA regimen. Therefore, we have decided to remove those statements, and replaced with “There are scarce evidence of efficacy and safety of different vaccine platforms in MHD patients since most of the clinical trials were using homologous mRNA vaccine regimen.” and “In addition, there are scarce evidence of efficacy and safety of different vaccine platforms in MHD patients since most of the clinical trials were using homologous mRNA vaccine regimen. Therefore, data on the currently available vaccines against SARS-CoV-2 infection, in terms of efficacy and safety of various vaccine regimen among MHD patients, is still limited.”

“There are scarce evidence of efficacy and safety of different vaccine platforms in MHD patients since most of the clinical trials were using homologous mRNA vaccine regimen.”

Abstract
Background
Page 1, paragraph 1, line 24-26

“In addition, there are scarce evidence of efficacy and safety of different vaccine platforms in MHD patients since most of the clinical trials were using homologous mRNA vaccine regimen. Therefore, data on the currently available vaccines against SARS-CoV-2 infection, in terms of efficacy and safety of various vaccine regimen among MHD patients, is still limited.”

Introduction
Page 2, paragraph 2, line 58-62

Moreover this manuscript does not address this question in not having healthy controls as a comparator. 

Answer: According to the Thailand’s Public Health Policy, MHD patients is one of the high-risk populations which had prioritized to receive SARS-CoV-2 vaccination. Hence, the healthy controls which are matching with our populations are difficult to collect. We have added this limitation in discussion part.

“Nevertheless, our study did have some limitations, including the fact that there were a relatively small number of MHD patients included in each vaccine regimen and a lack of healthy controls.”

Discussion
Page 12, paragraph 4, line 372-374

Safety and efficacy of different vaccines and vaccine combinations has been covered in the published literature and these various studies should have been referenced. 

Answer: We would like to thank the reviewer for your suggestion. We have added some references as suggested by the reviewer.

“Growing amounts of evidence have indicated that the heterologous prime-boost regimen could promote a higher degree of immunogenicity during a shortening of the time interval between injections under the safety profile [27-30]”

“Our results were similar to most of the previous studies of each vaccine regimens among general population [8,32].”

Reference
8. Tanriover, M.D.; DoÄŸanay, H.L.; Akova, M.; Güner, H.R.; Azap, A.; Akhan, S.; Köse, Åž.; Erdinç, F.; Akalın, E.H.; Tabak Ö, F.; et al. Efficacy and safety of an inactivated whole-virion SARS-CoV-2 vaccine (CoronaVac): interim results of a double-blind, randomised, placebo-controlled, phase 3 trial in Turkey. Lancet 2021, 398, 213-222, doi:10.1016/s0140-6736(21)01429-x.
27. Borobia, A.M.; Carcas, A.J.; Pérez-Olmeda, M.; Castaño, L.; Bertran, M.J.; García-Pérez, J.; Campins, M.; Portolés, A.; González-Pérez, M.; García Morales, M.T.; et al. Immunogenicity and reactogenicity of BNT162b2 booster in ChAdOx1-S-primed participants (CombiVacS): a multicentre, open-label, randomised, controlled, phase 2 trial. Lancet 2021, 398, 121-130, doi:https://doi.org/10.1016/S0140-6736(21)01420-3.
28. Tenbusch, M.; Schumacher, S.; Vogel, E.; Priller, A.; Held, J.; Steininger, P.; Beileke, S.; Irrgang, P.; Brockhoff, R.; Salmanton-García, J.; et al. Heterologous prime–boost vaccination with ChAdOx1 nCoV-19 and BNT162b2. Lancet Infect. Dis. 2021, 21, 1212-1213, doi:https://doi.org/10.1016/S1473-3099(21)00420-5.
29. Hillus, D.; Schwarz, T.; Tober-Lau, P.; Vanshylla, K.; Hastor, H.; Thibeault, C.; Jentzsch, S.; Helbig, E.T.; Lippert, L.J.; Tscheak, P.; et al. Safety, reactogenicity, and immunogenicity of homologous and heterologous prime-boost immunisation with ChAdOx1 nCoV-19 and BNT162b2: a prospective cohort study. Lancet Respir. Med. 2021, 9, 1255-1265, doi:https://doi.org/10.1016/S2213-2600(21)00357-X.
30. Yorsaeng, R.; Vichaiwattana, P.; Klinfueng, S.; Wongsrisang, L.; Sudhinaraset, N.; Vongpunsawad, S.; Poovorawan, Y. Immune response elicited from heterologous SARS-CoV-2 vaccination: Sinovac (CoronaVac) followed by AstraZeneca (Vaxzevria). medRxiv 2021, 2021.2009.2001.21262955, doi:10.1101/2021.09.01.21262955.
32. Falsey, A.R.; Sobieszczyk, M.E.; Hirsch, I.; Sproule, S.; Robb, M.L.; Corey, L.; Neuzil, K.M.; Hahn, W.; Hunt, J.; Mulligan, M.J.; et al. Phase 3 Safety and Efficacy of AZD1222 (ChAdOx1 nCoV-19) Covid-19 Vaccine. The New England journal of medicine 2021, 385, 2348-2360, doi:10.1056/NEJMoa2105290

Discussion
Page 12, paragraph 3, line 354-355

What sets this aside is the comparison of different vaccine sequences, which is covered more extensively here than in other publications.  Assuming that there are no differences in vaccine sequences due for clinical reasons, the data could have value if there were a healthy comparator group and adequate rationale for studying the different vaccine combinations were provided.

Answer: We would like to thank the reviewer for your compliment and suggestion. We agree that additional data of vaccine sequences, different vaccine combinations, and a healthy comparator group could enhance the value of this publication. However, the hypothesis of vaccine sequences, and different vaccine combinations on the clinical reasons are limited to assess, since Thailand’s Public Health Policy was strictly assigned the sequence of the vaccine and there was limited vaccine availability on that emergent situation. Therefore, neither MHD patients nor healthy comparator group is received other vaccine regimens apart from those mentioned in our clinical study.

Thank you again for all the assistance given by both editor and reviewers. Following extensive amendments in line with your very helpful comments we wish to resubmit out work for your consideration for publication in the MDPI Vaccines.

We look forward to your response.

Yours sincerely,

Sincerely yours,

Vuddhidej Ophascharoensuk (signed)

Vuddhidej Ophascharoensuk, MD.

Division of Nephrology, Department of Internal medicine, Faculty of Medicine,

Chiang Mai University, Chiang Mai, Thailand.

Tel. +66 53 935482    Fax: +66 53935481    

Email: vuddhidej.o@cmu.ac.th

ORCID: 0000-0002-6041-7989

  1. Authors and affiliations:

-Please replaced Dr. Karn Pongsuwan, M.D. to Dr. Karn Pogsuwan, M.D.

-Please changed the E-mail address of Dr. Romanee Chaiwarith, M.D. (E-mail: romanee.c@cmu.ac.th) and Dr. Poramed Winichakoon, M.D. (E-mail: poramed.wi@cmu.ac.th)

  1. Word count: 201 words (Abstract), 3,364 words (Introduction to Conclusion), 16 pages, 4 Figures, 4 Tables, 3 Supplementary Tables, and 33 References.

Round 2

Reviewer 2 Report

The limitation that healthy controls were not available remains.  However there is data suggesting that the heterologous prime boost is superior to a homologous regiment with the vaccines tested. 

To more accurately present one of the key statements of the manuscript I would suggest the following:

Lines 24-25, 54-56   “There is scarce evidence of the efficacy and safety of different vaccine prime-boost combinations in MHD patients since most clinical trials have used homologous mRNA vaccine regimens.

Author Response

Quincy Zhang

Editor-in-Chief

Vaccines

Re: “Immunogenicity and Safety of Homologous and Heterologous Prime-Boost of CoronaVac® and ChAdOx1 nCoV-19 Among Hemodialysis Patients: An Observational Prospective Cohort Study” (Reference number: Vaccines-2243165)  

March 16rd, 2023

Dear Editors of Vaccines

Enclosed please find the revised manuscript entitled “Immunogenicity and Safety of Homologous and Heterologous Prime-Boost of CoronaVac® and ChAdOx1 nCoV-19 Among Hemodialysis Patients: An Observational Prospective Cohort Study” for consideration for publication in the Vaccines. We would like to thank the editors and the reviewers for your valuable suggestions and the opportunity to revise and improve our manuscript.

We have revised our manuscript in accordance with the comments and suggestions made. Please find the responses to the reviewer and the revisions made to the manuscript below.

Responses to the Editor and the Reviewers

Responses to Reviewer #2

The limitation that healthy controls were not available remains.  However, there is data suggesting that the heterologous prime boost is superior to a homologous regiment with the vaccines tested.

To more accurately present one of the key statements of the manuscript I would suggest the following:

Lines 24-25, 54-56   “There is scarce evidence of the efficacy and safety of different vaccine prime-boost combinations in MHD patients since most clinical trials have used homologous mRNA vaccine regimens.

Answer: Thank you very much for your suggestion. We have added the suggested statement in our manuscript.

There is scarce evidence of the efficacy and safety of different vaccine prime-boost combinations in MHD patients since most clinical trials have used homologous mRNA vaccine regimens.”

Abstract

Background

Page 1, paragraph 1, line 24-26

“In addition, there is scarce evidence of the efficacy and safety of different vaccine prime-boost combinations in MHD patients since most clinical trials have used homologous mRNA vaccine regimens.”

Introduction

Page 1, paragraph 2, line 60-62

Thank you again for all the assistance given by both editor and reviewers. Following extensive amendments in line with your very helpful comments we wish to resubmit out work for your consideration for publication in the MDPI Vaccines.

We look forward to your response.

Yours sincerely,

Sincerely yours,

Vuddhidej Ophascharoensuk (signed)

Vuddhidej Ophascharoensuk, MD.

Division of Nephrology, Department of Internal medicine, Faculty of Medicine,

Chiang Mai University, Chiang Mai, Thailand.

Tel. +66 53 935482    Fax: +66 53935481    

Email: vuddhidej.o@cmu.ac.th

ORCID: 0000-0002-6041-7989

  1. Word count: 201 words (Abstract), 3,365 words (Introduction to Conclusion), 16 pages, 4 Figures, 4 Tables, 3 Supplementary Tables, and 33 References.
